# Engineering the vibrational coherence of vision into a synthetic molecular device

Moussa Gueye[1], Madushanka Manathunga[2], Damianos Agathangelou[1], Yoelvis Orozco[1], Marco Paolino [3], Stefania Fusi[3], Stefan Haacke[1], Massimo Olivucci [1,2,3] & Jérémie Léonard[1]

The light-induced double-bond isomerization of the visual pigment rhodopsin operates a molecular-level optomechanical energy transduction, which triggers a crucial protein structure change. In fact, rhodopsin isomerization occurs according to a unique, ultrafast mechanism that preserves mode-specific vibrational coherence all the way from the reactant excited state to the primary photoproduct ground state. The engineering of such an energy-funnelling function in synthetic compounds would pave the way towards biomimetic molecular machines capable of achieving optimum light-to-mechanical energy conversion. Here we use resonance and off-resonance vibrational coherence spectroscopy to demonstrate that a rhodopsin-like isomerization operates in a biomimetic molecular switch in solution. Furthermore, by using quantum chemical simulations, we show why the observed coherent nuclear motion critically depends on minor chemical modifications capable to induce specific geometric and electronic effects. This finding provides a strategy for engineering vibrationally coherent motions in other synthetic systems.

[1] Université de Strasbourg, CNRS, Institut de Physique et Chimie des Matériaux de Strasbourg, UMR 7504, F-67034 Strasbourg, France. [2] Department of Chemistry, Bowling Green State University, Bowling Green, OH 43403, USA. [3] Dipartimento di Biotecnologie, Chimica e Farmacia, Università di Siena I-53100 Siena, Italy. Correspondence and requests for materials should be addressed to M.O. (email: molivuc@bgsu.edu) or to J.L. (email: Jeremie.Leonard@ipcms.unistra.fr)

hile quantum mechanics rules chemical structure and reactivity, the field of quantum biology raises the question whether it may also rule biological functions like, for instance, in photosynthetic light-harvesting complexes possibly taking advantage of quantum coherence to enhance the efficiency of photochemical energy transfer and conversion[1–3]. A paradigmatic system at the interface of quantum chemistry and quantum biology is the animal visual pigment rhodopsin (Rh), in which the protein scaffold optimizes the photoisomerization of its co-factor, a protonated Schiff base of retinal (PSBR, see Fig. 1a). Indeed, the PSBR of Rh undergoes a high speed (200 fs[4]) and high quantum yield ($\Phi$)[5] isomerization, initiating the protein's biological function. This event is driven by the vibrationally coherent nuclear motion of the chromophore along a barrierless excited state ($S_1$) potential energy surface (PES) leading to decay to the ground state ($S_0$) in the region of a conical intersection (CInt)[6–11]. Previous studies argued that the high $\Phi$ value of Rh is achieved through a precise phase relationship between two vibrational modes at the point of decay[12]. A vibrationally coherent motion would propagate this phase relationship from the Franck–Condon (FC) state to the decay region, thus providing a way of controlling the $\Phi$ value. Therefore, replicating such a mechanism in synthetic molecules would provide a route for the preparation of molecular devices with properties and efficiency programmed at the atomic level.

In the past, the above idea has stimulated mixed theoretical/experimental research efforts. Following quantum chemical modeling, the N-alkylated indanylidene–pyrroline (NAIP) molecular skeleton was synthesized and functionalized such that, in solution, it mimics the $\pi$-electron system and geometrical constraints of PSBR in rhodopsins[13]. As a result, the MeO-NAIP (see structure 1 in Fig. 1b) was observed to undergo an ultrafast photoisomerization[14] with transient absorption spectroscopy data displaying low-frequency (60 to 80 cm$^{-1}$, i.e., ~500 fs period) oscillatory features[15,16] similar to those of the visual pigment featuring a 11-cis PSBR or light-sensing pigments featuring a 13-cis PSBR[17]. On the basis of quantum chemical simulations, these oscillations were attributed to ring deformation motion coupled to the reactive C=C bond twisting in the photoproduct[16], indicating a coherent nuclear motion initiated in $S_1$ and continued in $S_0$ after decay through a CInt. However, as we will further discuss below, resonant laser pulses may trigger coherent vibrational motion also in the initial $S_0$ state. For this reason, the proposed coherent $S_1$ decay of 1 remains hypothetical. In addition, the photoisomerization quantum yield of 1 is significantly lower[13,14] than that of Rh, and the precise understanding of the $S_1$ dynamics is an unavoidable pre-requisite for the development of more efficient NAIP-based switches.

Here we perform UV-VIS transient absorption (TA) experiments with sub-10 fs pump pulses and broadband white light probing to perform vibrational coherence spectroscopy[18] (see Methods section), in an experimental approach similar to that[19] recently applied to Rh[9]. When it is resonant with any molecule's absorption band, such a short pump pulse impulsively produces a non-stationary population described as a vibrational wavapacket in $S_1$. It may, however, also produce a vibrational wavepacket in $S_0$ via the so-called Impulsive-Stimulated Raman Scattering (ISRS) process[20–25]. This mechanism still operates with an off-resonant pump pulse, which produces vibrational wavepackets in $S_0$ only. Here we compare the effects of resonant and off-resonance excitations on 1 and 2, where the methyl group on carbon C5 has been replaced by a hydrogen atom (see Fig. 1c). As we will detail below, such a comparison provides a compelling evidence that: (i) the $S_0$ low-frequency mode, dominating the observed vibrational coherence of 1, originates in $S_1$ and (ii) an elementary chemical modification of the MeO-NAIP structure, affecting its torsional geometry, quenches the signatures of such critical $S_1$ motion in 2.

## Results

**Vibrational coherence spectroscopy**. The TA data obtained upon resonant excitation of 1 and 2 are displayed in Fig. 2a, b, respectively. They may be interpreted, via the introduction of an effective linear susceptibility[23], as the time-dependent, linear absorption of the probe beam by the non-stationary states impulsively produced by the pump pulse in $S_0$ and $S_1$. Accordingly, they reveal simultaneously transient species population kinetics in the form of UV-Vis absorption (from $S_1$, bleached $S_0$, or photoproduct) or emission (from $S_1$ only), as well as the accompanying vibrational dynamics in terms of oscillations. We first describe the signatures of the electronic population kinetics. Negative signals are due to $S_0$ bleaching (GSB) observed at $\lambda < 400$ nm, or to stimulated emission (SE) at $\lambda > 450$ nm. Positive signals are due to $S_1$ absorption (ESA) at early times and absorption of the vibrationally hot $S_0$ photoproduct (PA). The main difference between the electronic population kinetics of 1 and 2 is seen in the SE spectral shape and lifetime as well as in the early photoproduct signal. In compound 1 the SE extends far to the red (>700 nm). It impulsively decays to give rise to an early, almost octave-spanning, absorption spectrum of the

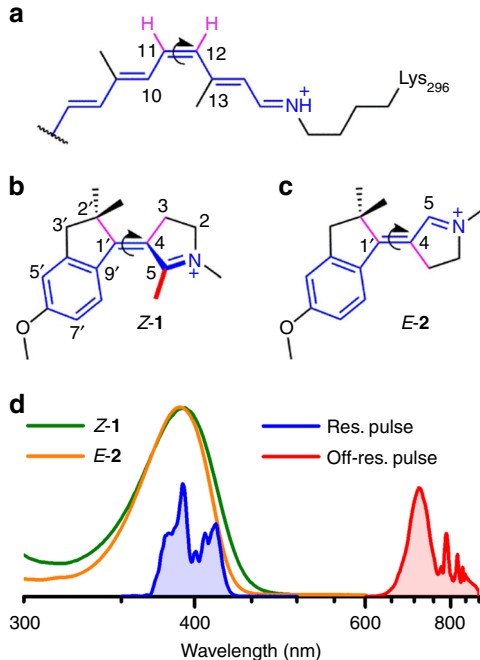

**Fig. 1** Structure and absorption spectra of the NAIP compounds. **a** The 11-cis PSBR chromophore of Rh (the curly arrow indicate the 11-cis to all-trans photoisomerization reaction) inspired the chemical design of (**b**) MeO-NAIP 1 and (**c**) "C5-demethylated" dMe-MeO-NAIP 2. Their most stable $S_0$ isomers are Z-1 and E-2. The elementary CH$_3$ to H substitution in Z-1 at position C5 alters the Z/E equilibrium in favor of a more stable E form. The three compounds have a similar $\pi$-electron system (blue molecular backbones) with an $S_1$ state characterized by a bond length alternation and partial reduction of the protonated Schiff base through charge transfer along the carbon backbone. They also have an analogous photoreaction coordinate, which involves the torsion around the central C=C double bond (curved arrows) as well as out-of-plane motions of the purple bonds. **d** Steady-state absorption spectra of Z-1 (in green) and E-2 (in orange) together with the femtosecond laser pulse spectra used for resonant (blue) and off-resonance (red) vibrational coherence spectroscopy

photoproduct (420–700 nm at 300 fs). The same spectral signatures have already been reported for a set of NAIP compounds[15,16] as well as for Rh[7] and were attributed to a vibrational wavepacket evolving coherently through the CInt and driving the impulsive decay from $S_1$ to $S_0$. The signal zero crossing at the crossover between SE and PA in the low-energy side of the spectrum, occurring at 250 +/− 30 fs at 700 nm for **1**, is thus interpreted as a measure of the time at which the molecular system crosses the CInt. In contrast, no such red-shifted SE and PA signatures are observed for **2**. More specifically, the SE is spectrally narrower, longer-lived and the narrow PA signature rises progressively during the SE decay, both according to exponential kinetics (ref. [26] and Supplementary Information of ref. [27]), in contrast to the TA signals of **1**, which show dominant non-exponential features due to vibrational wavepackets.

The oscillatory signals (Fig. 2e, f, g, h) are isolated from the underlying electronic population kinetics by globally fitting the latter with a sum of exponential decay functions[18]. Fourier transforming the fit residuals reveals the frequencies associated with the vibrational wave packets observed along a time window covering the $S_1$ reactive motion, decay, and $S_0$ photoproduct

formation. Figure 3a, b display the corresponding vibrational power spectra in blue. When using the non-resonant red to IR pulse instead of the 400 nm pulse as a pump pulse, no population is created in $S_1$ and no electronic population dynamics is detected. However, coherent vibrational dynamics are excited in $S_0$, by non-resonant ISRS, producing oscillatory TA signals, which are directly analyzed by Fourier transformation. The corresponding vibrational spectra are displayed in red in Fig. 3a, b.

For both compounds **1** and **2**, the high-frequency vibrational activity (>500 cm⁻¹) is observed in both on- and off- resonance experiments, and is therefore attributed to $S_0$ vibrational activity. The dominating signatures are the 1572 cm⁻¹ ethylenic stretch mode, the 1259 cm⁻¹ mode localized on the indanylidene moiety[28] as well as the 734 cm⁻¹ and 651 cm⁻¹ (**2** only) modes reported here for the first time and unassigned. The 455 cm⁻¹ mode (unassigned) may also be an $S_0$ mode but its detection upon off-resonant excitation is likely impaired by the much more intense 492 cm⁻¹ mode of the fused silica sample cuvette. In contrast, for both compounds, the low-frequency vibrational activities (<400 cm⁻¹) observed upon resonant excitation (blue spectra, Fig. 3a, b) are not detected in the off-resonant

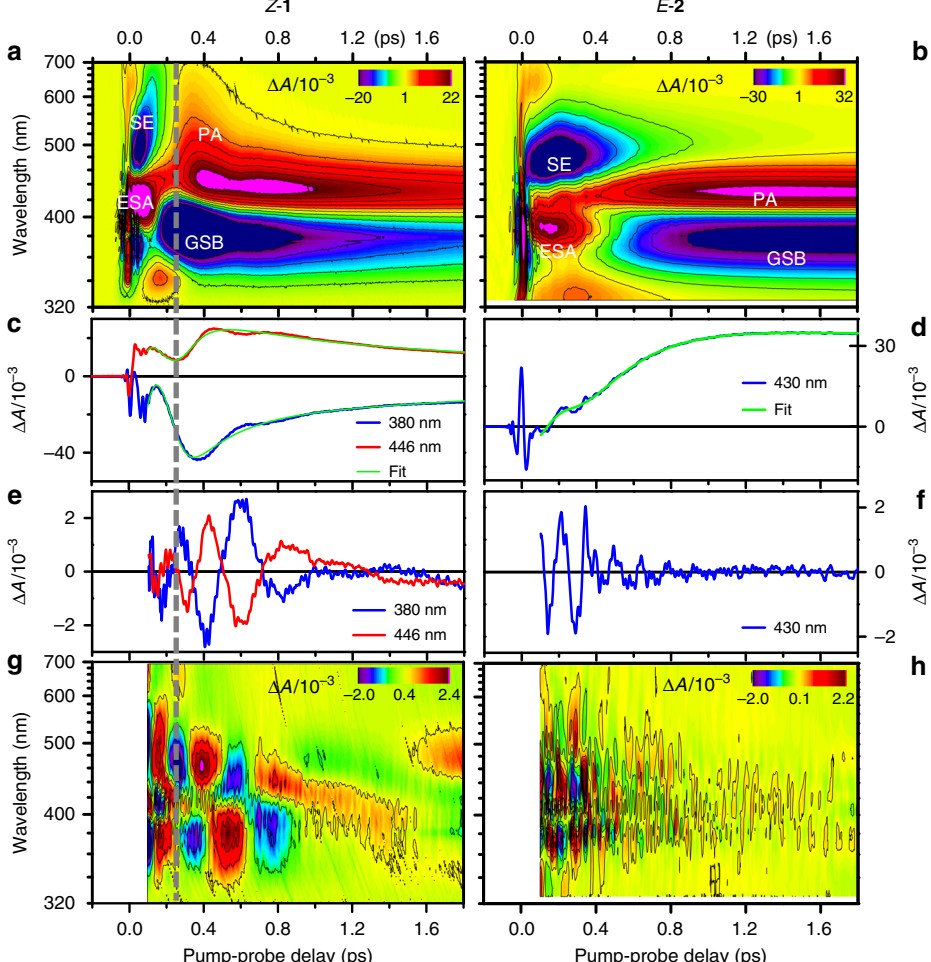

**Fig. 2** Transient absorption spectroscopy of Z-**1** and E-**2** upon resonant excitation. **a, b** 2D map representations of the differential absorption (Δ*A*) coded in false colors, as a function of probing wavelength (nm) and pump-probe time delay (ps). The TA data are obtained upon excitation with a resonant, 8.5 fs pump pulse of a methanol solution of Z-**1** (left column) and E-**2** (right column). **c** Kinetic traces illustrating the signal detected at the 380 nm (blue trace) and 446 nm (red trace) probe wavelengths for Z-**1**, and the result of the 2D map global fit (green) at the same wavelengths. **d** Same for E-**2**, at the 430 nm probing wavelength. **e, f** The corresponding residuals reveal the oscillatory signatures of the nuclear motions. **g, h** 2D map representation of the residuals of the global fit of both TA data sets. Time delays shorter than 0.1 ps are disregarded (see Methods section). Fourier transformation of these maps along the time axis reveals the power spectra of the oscillatory signals, interpreted as vibrational spectra (Fig. 3). The vertical dashed line at 250 fs across **a**, **c**, **e**, **g** indicates the moment of impulsive $S_1$ decay of **1** at the CInt

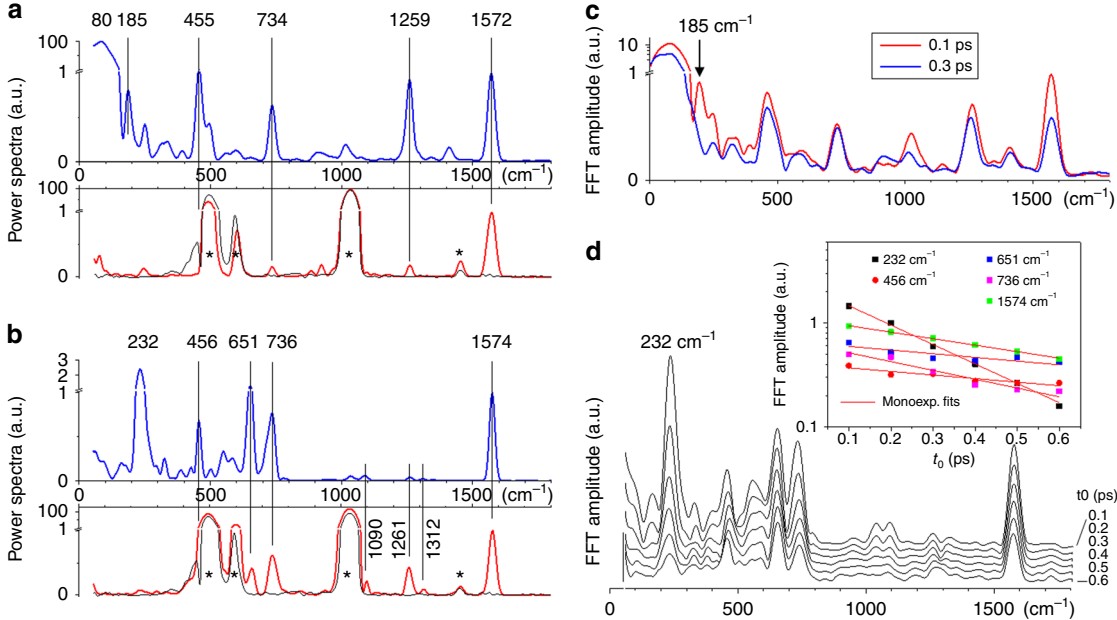

**Fig. 3** Vibrational coherence spectroscopy of compounds *Z*-**1** and *E*-**2**. The power spectra of the differential absorption Δ*A* oscillatory residuals revealed in Fig. 2g, h are averaged over the 350–470 nm probing window and displayed in **a** for *Z*-**1** and **b** for *E*-**2**. The spectra are normalized to 1 at the C=C stretch frequency of 1572 or 1574 cm⁻¹. The vertical scale is linear from 0 to 1 and logarithmic above 1. They reveal the frequencies of the modes in which vibrational wave packets are observed upon resonant excitation at 400 nm (blue spectra). The same analysis is performed on the data acquired upon off-resonance excitation at 800 nm (red spectra). The black spectra correspond to off-resonant excitation of the pure methanol solvent. Stars indicate the vibrational signatures of the solvent or fused silica cuvette. **c** The same Fourier analysis of the *Z*-**1** oscillatory residuals is performed now on a sliding, 1-ps long time window starting at $t_0 = 0.1$ ps (red spectrum) or $t_0 = 0.3$ ps (blue spectrum) i.e., respectively before and after impulsive decay. The dominant 80 cm⁻¹ mode persists. The 185 cm⁻¹ feature instead disappears upon decay to S₀, but it seems too narrow to represent an oscillatory signal, which would last for only 250 fs, which questions its significance and interpretation, especially in the proximity of the very intense 80 cm⁻¹ mode. **d** Same as **c**, for *E*-**2**, with $t_0$ starting times ranging from 0.1 to 0.6. The inset displays the semilog plot of the FFT amplitude of the five dominating modes at 232, 456, 651, 736, 1564 cm⁻¹, as a function of $t_0$, and the corresponding monoexponential decay fits. While the amplitude of all higher-frequency modes decay on the 600 to 1200 fs time scale, the damping of the 232 cm⁻¹ mode is observed to occur on a significantly faster 230 fs time scale, in line with the 300 fs excited state lifetime of *E*-**2**

experiments (red spectra, Fig. 3a, b). This observation is central and may be rationalized as follows. The ISRS mechanism at work here as well as spontaneous Raman scattering may both be described by a wavepacket formalism[25,29,30] which introduces the propagation time τ of the wavepacket on the S₁ PES between the two light field interactions that characterize a Raman transition. When τ is much shorter than a given vibrational period, no evolution occurs on S₁, such that no Raman activity exists for this mode unless non-Condon effects (i.e., nuclear coordinate dependence of the electronic transition dipole moment) become significant. This conclusion holds for non-resonant Raman processes, where $\tau \sim h/\Delta E$ with Δ*E* the detuning of the light field from resonance[30] (here, with the 800-nm pump, Δ*E* ~ 1.5 eV and τ ~ 2.7 fs). The same conclusion also holds for resonant ISRS performed with a short enough pump pulse[31], since in this case, τ is of order of the pump pulse duration (8 fs here). Therefore, we argue that for sufficiently low-frequency modes (i.e. oscillation periods much longer than 8 fs) the vibrational activity induced in S₀ by ISRS is the same for both on- and off-resonance pulses (and results from non-Condon effects, if any). In fact, the analysis of the S₁ reaction coordinate discussed below confirms that only a high-frequency stretching mode is activated within 8 fs, while the isomerization motion is activated at a later time.

Of central interest, here is the 80 cm⁻¹ mode, which largely dominates the vibrational power spectrum of **1** in the resonant experiment only (2 orders of magnitude more intense than the 1572 cm⁻¹ mode; notice the break in vertical scale in Fig. 3a, b). This mode actually corresponds to the low-frequency oscillation

readily observed in **1** in Fig. 2e, g up to the 1 ps time scale, i.e., after the impulsive S₁ decay as also confirmed by the analysis in Fig. 3c. In previous TA experiments on **1** with a resonant, 80 fs, pump pulse, the same oscillation was also observed with similar amplitude relative to the overall TA signal, and attributed to a coherent nuclear motion in S₀[15,16]. The present comparison between on- and off-resonant excitation (with pulse durations now much shorter than the ca. 400 fs period of the vibrational mode) demonstrates that the 80 cm⁻¹ oscillation observed in **1** is not due to ISRS, but results from the reactive motion of the population initially promoted to S₁. As a corollary, the vibrational coherence initiated on S₁ along this mode is preserved upon decay to S₀. Remarkably, in *E*-**2**, the 80 cm⁻¹ mode is not detected. Instead an intense vibrational activity is observed at 232 cm⁻¹. However, it is quenched upon decay to S₀ as illustrated in Fig. 3d meaning that this is the signature of an S₁ vibrational coherence which is not transferred to S₀.

**Quantum chemical modeling and mechanistic interpretation.** The above experimental data lead us to the following conclusions: (i) the S₀ 80 cm⁻¹ mode of compound **1** is activated exclusively via S₁ coherent nuclear motion and the coherence is preserved upon impulsive decay through the CInt. (ii) The removal of the methyl group on C5 quenches the impulsive S₁ decay as well as the signatures of vibrational coherence spanning both the S₁ and S₀ PESs. We now provide a mechanistic interpretation of these points. Firstly, both crystallographic (see Supplementary Figs 1

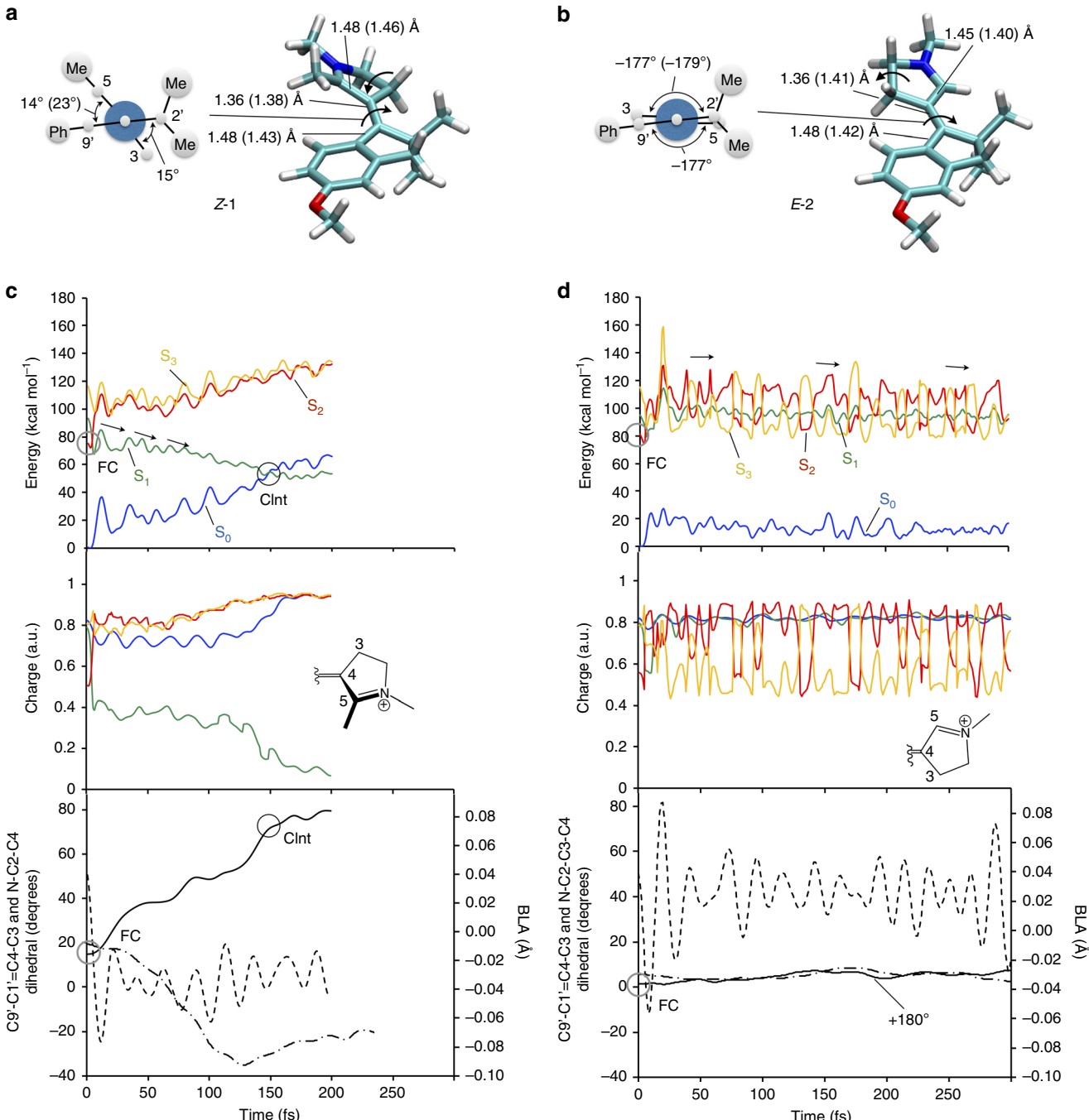

**Fig. 4** Mechanistic interpretation of the influence of the methyl substitution on C5. **a**, **b** Structure of the computed $S_0$ free energy minima of Z-**1** and E-**2** in methanol at room temperature and values of their relevant structural parameters. The values for the corresponding $S_1$ energy minima are given in parenthesis and the values of the C9′–C1′=C4–C3 and C2′–C1′=C4–C3 torsional parameters are given in the Newman projections on the left. A comparison with the available observed crystallographic parameters is given in Supplementary Fig. 2. **c**, **d** Computed $S_1$ trajectories of Z-**1** and E-**2**, respectively, in methanol solution, illustrated by (top panels) the $S_0$, $S_1$, and $S_2$ CASPT2//CASSCF/6-31G*/Amber energy profiles, (middle panels) the changes in electronic structure along the trajectories in terms of the fractional positive charge remaining on the pyrroline moiety of the switch, and (bottom panels) the progression along the reaction coordinate described through skeletal double-bond stretching (BLA, dashed line), out-of-plane deformation of the pyrroline ring (reflected by the =N–C2–C3–C4– dihedral, dashed-dotted line), double-bond twisting (C9′–C1′=C4–C3, full line). FC indicates the configurations of **a** and **b**. CInt indicates a conical intersection point. Notice the slow progression of E-**2** along both twisting and out-of-plane deformation coordinates

and 2) and computed room-temperature structures in methanol solution (see Supplementary Notes 1 and 2, and Supplementary Figs 3–5) show that the reactive C1′=C4 bond is pre-twisted in Z-**1** but substantially planar in E-**2**, as illustrated in Fig 4a, b. Secondly, the nuclear motion initiated in the spectroscopically

allowed $S_1$ state of Z-**1** and E-**2** is dramatically different. Indeed, as displayed in Fig. 4c, the energy profiles along the FC trajectory of Z-**1** show that, similar to 11-*cis* PSBR in Rh[11], the system evolves along a steep $S_1$ PES reaching a CInt rapidly. More precisely, the vibrational wavepacket, whose center is assumed to

move along the FC trajectory (see Methods section), would decay along a segment of the intersection space containing an infinite number of Cint's[32]. In contrast with $Z$-**1**, $E$-**2** shows, after a fast initial relaxation, a progression along flat and quasi-degenerate PESs consistently with an ineffective stretching-torsion coupling (Fig. 4d and Supplementary Figs 6–8). This behavior is consistent with the $S_1/S_2$ crossing detected along the PES for $E$-**2** but not for $Z$-**1** (see Supplementary Note 3 and Supplementary Figs 6 and 7). The same electronic effect was documented in a comparative study of Rh and microbial rhodopsins[33], where the longer $S_1$ lifetime of microbial rhodopsins is attributed to a crossing and re-crossing of nearly degenerate $S_1$ and $S_2$ states. It is thus apparent that the steric hindrance and the pre-twisting introduced by the C5-substituent induce the following: (i) a dominating $Z$ configuration in $S_0$ and (ii) a separation of the $S_1$, $S_2$ and $S_3$ states. Also, a charge transfer character analog to the visual pigment is observed in $Z$-**1**, whereas in $E$-**2** the intertwining $S_1$, $S_2$ and $S_3$ states produces an oscillating electronic character modulated by the ethylenic stretch (see Fig. 4d and Supplementary Fig. 6).

The analogy between synthetic and biological systems can also be extended to the $S_1$ reaction coordinate. According to our calculations (see Fig. 4c, d bottom panels as well as Supplementary Figs 8 and 9), within the first 10 fs, only a high-frequency stretching mode is activated. Then, in **1** only, large out-of-plane ring deformations coupled to the central bond twisting are activated, which result in the rotation of the pyrrolinium ring relative to the indanone moiety. Such a coordinate is responsible for the effective breaking of the $\pi$-bond at the electronic level and parallels the one documented for Rh[12]. More specifically, the NAIP C1′=C4 twisting is straightforwardly associated with the C11=C12 twisting of the Rh chromophore, while the NAIP five-membered ring inversions, i.e., C2′ and C3 carbon-out-of-plane motions, mimic the hydrogen-out-of-plane motions of the HC11=C12H moiety of the Rh chromophore (Fig. 1a, b). The reaction coordinate will continue after decay to $S_0$[16], leading to the coherent population of the prominent $80 \, \text{cm}^{-1}$ mode demonstrated here. Whether or not vibrational coherence is also preserved along the ring inversion motions remains speculative, since the corresponding sign of vibrational activity, expected in the $200-350 \, \text{cm}^{-1}$ range (also not activated in the off-resonance experiments), is possibly activated by the $S_1$ motion in **1**, but remains close to the noise level in the present experiment (Fig. 3c).

## Discussion
In conclusion, by using vibrational coherence spectroscopy and quantum chemical simulations, we have shown that the $S_1$ force field of $Z$-**1** triggers a ballistic reactive motion towards the CInt and decay to $S_0$. In a statistical ensemble of molecules in solution at room temperature, the observation of vibrational coherence in the photoproduct requires a degree of synchronization between all decay events in the ensemble, which demonstrates the ballistic motion and indicates that such motion is poorly affected by the initial nuclear velocities at ambient temperature. In contrast, $E$-**2** experiences a more diffusive motion towards the CInt. The corresponding loss of synchronization would then explain the observed loss of ensemble coherence. This interpretation leads to the hypothesis that while $Z$-**1** replicates the coherent dynamics that controls the Rh photochemistry, such a regime is not present in $E$-**2**, demonstrating that the observed biomimetic behavior can be switched off. Furthermore, the documented $Z$-**1** and $E$-**2** contrasting regimes (Fig. 5) provide a parallel between molecular switches and rhodopsin photoreceptors shedding new light on the mechanism by which Rh itself may optimize its photoisomerization.

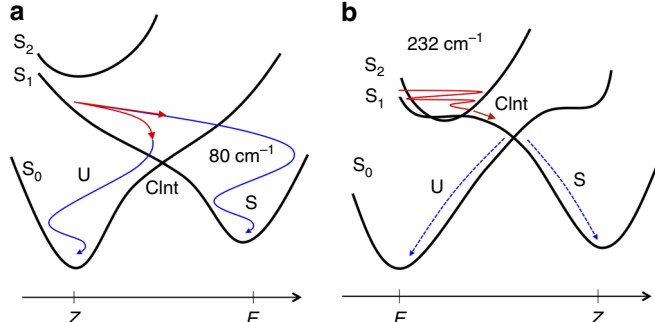

**Fig. 5** Coherent and incoherent isomerization motions in the CInt vicinity. **a**, **b** Schematic representations of the $S_0$, $S_1$ and $S_2$ PES's (black solid curves) of $Z$-**1** and $E$-**2**, respectively, with the illustration of the vibrationally coherent (solid colored lines) or incoherent (dashed colored lines) reactive motions in the $S_1$ (in red) or $S_0$ (in blue) PES's. Two decay channels corresponding respectively to successful (S) or aborted (U) isomerizations are shown

Apart from the two seminal examples of vibrationally coherent triiodide photodissociation[34] and Rh photoisomerization[6], unambiguous observation of reaction-induced vibrational coherences were reported in other photoreactions such as e.g., ultrafast low- to high-spin transition in an iron II molecular complex[35] or ring-opening in a molecular dyad[36]. However, in the latter two cases, the vibrational coherence was interpreted as being transferred to low-frequency modes not directly related to the reaction coordinate. Here we demonstrate that the onset (i.e., turning on or off) of a vibrationally coherent reactive motion can be induced via chemical modification and change in equilibrium geometry. This provides the opportunity to precisely control the decay event and, likely, to increase the presently low photoproduct quantum yield of NAIPs ($\Phi = 0.20\%$)[13,14] or other retinal derivatives. Indeed, as previously proposed[12], the conservation of a precise phase relationship between the HOOP and C11=C12 twisting modes of Rh would be responsible for its high $\Phi$ value ($\Phi = 67\%$)[5]. Thus, assuming a biomimetic behavior, a coherent motion would be a pre-requisite for achieving the optimal phase relationship between the C1′=C4 twisting and five-membered ring inversion in NAIPs and, in turn, for improving their photoisomerization efficiency. Notice, that this scenario is in contrast with the previous suggestion that $\Phi$ would correlate monotonously with the reaction speed according to the Landau-Zener model[6,17,37]. Finally, the control, via a chemical modification, of geometric and electronic effects appears a valid general strategy for coherently populating the torsional motion of other C=C isomerizing compounds.

## Methods
**Experimental**. Compounds **1** or **2** were synthesized as previously reported[13,26]. The isomer content of the samples in the dark at room temperature was determined to be >97% $Z$ for **1** and 95% $E$ for **2**, by prior $^1$H-NMR spectroscopy in deuterated methanol.

Vibrational coherence spectroscopy is performed by recording TA data with an experimental set-up described in detail in ref. [18]. In short, we use a 1 kHz amplified Ti:Sa laser system delivering 3 mJ, 40 fs, 800 nm pulses. About 1 mJ of the fundamental pulse is injected in a neon-filled hollow fiber (Imperial College Consultants) where non-linear, guided propagation induces spectral broadening and generates the structured, red to IR spectrum displayed in Fig. 1d. Subsequent pulse compression using chirped mirrors results in a ~0.5 mJ, 6 fs pulse used for off-resonant excitation. Type II sum frequency generation of this 6 fs pulse with the fundamental 800 nm pulse generates the 45-nm broad pulse centered at 400 nm and used for resonant excitation. The FWHM duration of the latter blue pulse is measured to be 8.5 fs by implementing 2D spectral shearing interferometry (2DSI) [38,39]. Before TA experiments, the duration of either pump pulse is optimized directly in the sample by fine tuning group velocity dispersion with a pair of fused silica wedges (in combination with the chirped mirrors) so as to generate a white

light supercontinuum in the solvent (methanol) with minimum pump intensity. Then the pump intensity is further reduced to 50 nJ per pulse to perform the actual experiments on the compounds.

As a probe pulse for TA spectroscopy we use a chirped, white light supercontinuum generated in $CaF_2$. Half the intensity of this pulse is used as a reference beam. Both probe and reference spectra are acquired with a prism spectrometer equipped with two CCD cameras operated at a 1-kHz acquisition frequency (commercial acquisition system by Entwicklungsbüro G. Stresing, Berlin). The reference spectrum is used to normalize the probe spectrum. Pump and probe beams are focused and overlapped in the liquid sample. The pump beam is chopped at 500 Hz, such that two successive probe spectra are used to compute the pump-induced absorption change, i.e., TA spectrum, of the sample. A 500-μm-travel piezoelectric transducer (PT) on the pump beam optical path is continuously oscillating at a 0.5-Hz period to scan the pump-probe delay. The instantaneous PT position is acquired in synchronicity with probe and reference spectra at 1 kHz, which allows us to assign a specific pump-probe delay to each probe spectra. The TA experiments are performed on 1.3 mM MeOH solutions of compounds **1** or **2** circulated with a peristaltic pump in a 0.2 mm-thick flow cell having 0.2-mm-thick fused silica windows (Hellma).

Before global fitting and fast Fourier transformation (FFT) of the residuals, the TA data sets are processed to correct for wavelength dependence of the time origin, induced by the chirp in the white light probe pulse. The signal recorded in the pure solvent is used to characterize the chirp for this purpose. The data at time delays shorter than 0.1 ps are disregarded in the analysis (global fit and FFT) because they are dominated by the pump-probe cross-modulation signal generated upon coherent interaction of both beams with the solvent. The oscillations (Fig. 2g, h) are most pronounced in the 350–470 nm probing window, resonant with the reactant ($S_0$), excited state ($S_1$), and photoproduct ($S_0$) absorptions.

**Computational**. The 300 K Boltzmann distribution of $Z$-**1** and $E$-**2** in methanol has been simulated by combining[40] the Average Solvent Electrostatic Configuration (ASEC) model[41] and the free energy gradient method proposed by Nagaoka et al.[42]. A representative Franck-Condon structure has then been obtained by energy minimization of the ASEC snapshot with the excitation energy closest to the average (Supplementary Notes 1 and 2, and Supplementary Figs 3–5[43]).

Excited state PES mapping (see Supplementary Fig. 6) and trajectory calculations have been performed using the CASPT2//CASSCF(12,11)/6-31G*/ Amber quantum mechanics/molecular mechanics protocol available through the MOLCAS-Tinker interface[44,45]. The results of Fig. 4 are confirmed at the XMCQDPT2 level (Supplementary Note 3 and Supplementary Fig. 7). The FC trajectories of **1** and **2**, i.e., trajectories that start from $S_0$ equilibrium geometries with zero initial velocities, are released from the computationally assigned spectroscopic state (see Supplementary Note 2 and Supplementary Fig. 5). As argued for other rhodopsin-like model compounds[46,47], and consistently with the PES scans (Supplementary Fig. 6), FC trajectories are assumed to represent the initial (i.e., within few hundred femtoseconds) excited state motion of the center of the vibrational wavepacket.

**Data availability**. The data that support the findings of this study are available from the corresponding authors upon reasonable request.

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

## Acknowledgements

We acknowledge the French Agence Nationale de la Recherche for support via grant No. ANR-11-JS04-010-01 "IPQCS", the "Investissement d'Avenir" program: Labex NIE ANR-11-LABX-0058_NIE, Labex CSC ANR-10-LABX-0026_CSC, Equipex UNION ANR-10-EQPX-52-01, the Région Alsace (Contrat doctoral, No. 607-12-C31), the Université de Strasbourg for a USIAS Fellowship, the NSF Grant No. CHE-1710191, and the Italian MIUR for grant PRIN 2015. We acknowledge the Ohio Supercomputer Center, the Mésocentre of the Université de Strasbourg and CINECA for granted computer time.

## Author contributions

M.G. and J.L. built the experimental set-up. M.G., D.A and J.L. acquired and analyzed the data. M.P. and S.F. synthesized the compounds. M.M. and Y.O. carried out the computational work. J.L., S.H. and M.O. wrote the paper.

## Additional information

**Competing interests:** The authors declare no competing financial interests

