## [Peer Review File · Nature Communications]

Reviewer #1 (Remarks to the Author):

In this manuscript, Léonard et al report on the photophysical study of a chromophore known as NAIP (N-alkylated indanylidene-pyrroline) which can be considered to mimic the photophysics of Rhodopsin chromophore, 11-cis retinal. A comparison is made between two structurally (slightly) different chromophores (a hydrogen atom replaces a methyl group in one version of the chromophore).

The experimental methods consisted of mainly time resolved vibrational coherent spectroscopy (sub-8 fs). But also a high level and detailed computational analysis of the chromophore photophysics was developed in this work.

The author's major claim is to have demonstrated that a rhodopsin-like isomerization operates in NAIP. Furthermore, the authors claim that the observed coherent nuclear motion associated with the isomerization process - sustained throughout the conical intersection leading to S1->S0 relaxation - critically depends on minor chemical modifications of the chromophore structure. The key observable in this work is provided by excited state vibrational modes sensitive to motion along the energy potential. By comparing resonant with non-resonant vibrational dynamics the authors were able to clearly assign vibrational modes only present in the excited state. The findings and claims are well supported by experimental and computational data and the originality of the work is significant. The results and analysis presented in this work are bound to influence thinking in this field and promote better understanding of how molecular mechanisms affect crucial protein structure changes.

I fully support the publication of this manuscript bar some improvements as follows.

- 1) A key observation is the maintenance of the coherent evolution of the vibrational mode at 80 cm⁻¹ during electronic relaxation from S1 energy level back to ground state through a conical intersection, whereas an adjacent mode at 185 cm⁻¹ is heavily quenched for Z-1. For E-2 the corresponding excited state mode at 232 cm⁻¹ smoothly loses amplitude as the electronic excited state relaxes back to the ground state. I believe this a key experimental observation which heavily supports the claim that for Z-1 the electronic relaxation towards the conical intersection happens ballistically whereas for E-2 this happens with a diffusive motion. Therefore this should be part of the main manuscript and not be relegated to SI. In fact Figure 5 could go to SI as it consists mostly of a cartoon representation of potential energy and does not add a huge amount of information.
- 2) Although Figure 4 conveys a great amount of information regarding the detailed computational analysis performed in this work, it is incredibly difficult to read. It has to be significantly simplified to more synthetically convey the key conclusions being drawn from it in the main manuscript.
- 3) The language with which one is suddenly faced at the start of the conclusion section does not fit with the more general, direct and clear language being used up to then and which makes the paper reading pleasant and understandable. For example, "the S1 force field of Z-1 transfers the system towards" and "to explain the observed loss of ensemble coherence". If this could be simplified, it could improve understanding of the paper significantly.
- 4) In the introduction the authors provide information about the quantum yield of the 11-cis retinal isomerization process, which I believe is a key parameter (and important information), and therefore I was expecting to get know how the biomimetic NAIP chromophore compares. But only in the conclusion this is mentioned. Maybe this could be brought up earlier and then more clearly make the connection of how the key findings in this paper can be used to design better structures with higher quantum yields?

Reviewer #2 (Remarks to the Author):

The paper reports a strategy for engineering vibrationally coherent motion in molecular systems that can operate an optomechanical energy transduction. For the study transient absorption measurement and quantum chemical simulations are performed. The topic is highly interesting and the paper is well written.

I have only some questions that should be addressed:

On page 2, second last paragraph: here the reference to "fig. 2 e) and f)" should be given explicitly instead of "Figure 2".

Is it clear that resonant and non-resonant excitations will induce the same dynamics in the electronic ground state?

Figure 1d: Please comment on the non-smooth laser pulse spectra

It did not become clear to me, also not from the SI, how many trajectories were actually run and evaluated to support the interpretation? Were they performed on CASPT2 level of theory or on CASSCF? In case they were performed on CASSCF some validation with respect to the CASPT2 results should be given.

Reviewer #3 (Remarks to the Author):

Gueye et al. studied ultrafast excited-state cis-trans isomerization of MeO-NAIP and deMe-MeO-NAIP in solution by time-resolved absorption spectroscopy employing sub-8 fs pump pulses for pumping. Owing to the ultrashort pump pulses whose duration is shorter than the vibrational periods of molecules, the authors succeeded in observing oscillatory features in the dispersed spectra of the probe pulse, which reflect coherent nuclear wavepacket motion induced with the pump process. Based on the comparison between the data obtained with resonant and non-resonant pump pulses, they found a few vibrations attributable to the excited-state and the ground-state photoproduct, although others are assignable to the vibration of the initial ground state which are induced by the impulsive Raman process. Interestingly, cis MeO-NAIP shows the stimulated emission and photoproduct absorption that indicate ballistic formation of the isomerized photoproduct, and for this molecule, they observed the reaction driven vibrational coherence in the photoproduct. In contrast, for deMe-MeO-NAIP which shows the stimulated emission and transient absorption representing usual populational conversion processes, they only observed excited-state vibrations that are induced by photoexcitation. With help of quantum chemical calculations, they discuss the difference in the ultrafast isomerization process of the two compounds, as well as its relevance to the prototypical ultrafast isomerization of rhodopsin.

Understanding nuclear wavepacket motion in the ultrafast chemical reactions is a key to elucidate the reactive multidimensional potential energy surface and how chemical reactions proceed on it. In particular, the coherent vibration that survives reactions is expected to provide some information about the transition state as well as the conical intersections, so that it is extremely intriguing. Therefore, this has been a central issue in physicochemical studies of chemical reactions for quite long, but it is only recently that it becomes possible to discuss in detail based on the results of ultrafast spectroscopy as well as quantum chemical computation. In this sense, the subject of this study is timely and the result is interesting. As the authors discuss in this paper, isomerization of rhodopsin is certainly a prototype of this problem, but the molecules they picked up are much simpler so that they may be more rigorously examined with quantum chemical calculations.

Because of the interesting problem that this study examined, I feel that this paper may be publishable. However, I have numbers of concerns. Thus, this should be published only if the authors can resolve the following concerns.

1.

The authors compared the data taken with resonant and non-resonant pump pulses for distinguishing the excited-state and ground-state vibrations. It is good, because assignments of numbers of early studies were done in a somewhat careless way, which generated substantial confusion. However, for the assignment of 80 cm⁻¹ vibration of MeO-NAIP, this is not enough because the selection rules of resonance Raman and non-resonant Raman are different. The former occurs with the Franck-Condon mechanism (A-term mechanism) and the latter obeys the

vibronic coupling mechanism (B-term mechanism). Therefore, the authors need to measure spontaneous resonant Raman spectra of the samples and confirm that the 80 cm⁻¹ band does not appear in the steady-state spectra. Recent advances in Raman spectroscopy allows us to readily measure low frequency vibrations, in particular for a mode having such a huge Raman cross section. Even in the case that spontaneous experiments are difficult (e.g. because of strong fluorescence), the discussion on the Franck-Condon activity based on the quantum chemical calculation must be made. It must be easy for the authors because the authors have already the results of computations for both of the S₀ and S₁ states. This is critically important for this paper because all the discussions rely on the assignment of this vibration.

2.

I noticed some of the authors' view are too simplified and/or too inaccurate. For example, first, the authors compared the isomerization of MeO-NAIP with that of rhodopsin rigorously, At least for me, however, MeO-NAIP looks a model of 13-cis although the retinal Schiff base of rhodopsin is 11-cis. Second, they discuss the difference between MeO-NAIP and diMe-MeO-NAIP by saying that "minor chemical modifications capable to induce specific electronic effects." However, the authors compared the results of "cis-form" of MeO-NAIP and the "trans-form" of diMe-MeO-NAIP. This is not just a change of methyl substitution but a big difference between the trans and cis forms. The authors must reconsider their discussion and rewrite their statements.

Minor points:

3.

For actual experiments that they performed, they just write "Following a method [20] recently applied to Rh [8] ... (page 2, para 2)" in the main text. It is impossible for readers to figure out what kind of actual measurements they carried out. Addition of relevant description is necessary.

4.

For explaining the spectra they show, the authors write "These data can be interpreted as a time-resolved linear absorption experiment ... (page 2, para 3)". They might be able to say so if the spectra only exhibit the excited-state transient spectra but, in this case, the stimulated emission, ground-state bleaching and the absorption of the photoproduct appear in the transient spectra. Therefore, such a too-simplified description will confuse the readers, although the authors may have tried to explain their observation to non-experts in a very easy manner. This part must be rewritten, or deleted.

5.

The vertical axis of the spectra in Figure 3 is very confusing. The authors mix the linear scale and the log scale without any explanation. They have to clearly mention this issue in the main text as well as in the figure caption.

6.

In page 4, para 2, the authors write, "Reaction-induced vibrational coherence were previously observed in a variety of ultrafast photoreactions". They have to explicitly mention and explain several examples, carefully choosing the literatures. In my opinions, it was "claimed" in a variety of ultrafast photoreactions, but there are not so many which obtained the consensus of the community.

7.

In discussion part, the authors discussed their observation with the conical intersection. The conical intersection is very important concept but is often misunderstood, in my opinion. In fact, the new term "conical seam" has been introduced because the conical intersection is not one point on the multidimensional potential energy surface. Since the theoretician who knows this issue very well is in the authors, I suggest mentioning this issue in this paper.

Answers to the referees

Reviewer #1 (Remarks to the Author):

In this manuscript, Léonard et al report on the photophysical study of a chromophore known as NAIP (N-alkylated indanylidene-pyrroline) which can be considered to mimic the photophysics of Rhodopsin chromophore, 11-cis retinal. A comparison is made between two structurally (slightly) different chromophores (a hydrogen atom replaces a methyl group in one version of the chromophore).

The experimental methods consisted of mainly time resolved vibrational coherent spectroscopy (sub-8 fs). But also a high level and detailed computational analysis of the chromophore photophysics was developed in this work.

The author's major claim is to have demonstrated that a rhodopsin-like isomerization operates in NAIP. Furthermore, the authors claim that the observed coherent nuclear motion associated with the isomerization process - sustained throughout the conical intersection leading to S1->S0 relaxation - critically depends on minor chemical modifications of the chromophore structure.

The key observable in this work is provided by excited state vibrational modes sensitive to motion along the energy potential. By comparing resonant with non-resonant vibrational dynamics the authors were able to clearly assign vibrational modes only present in the excited state.

The findings and claims are well supported by experimental and computational data and the originality of the work is significant. The results and analysis presented in this work are bound to influence thinking in this field and promote better understanding of how molecular mechanisms affect crucial protein structure changes.

I fully support the publication of this manuscript bar some improvements as follows.

1) A key observation is the maintenance of the coherent evolution of the vibrational mode at 80 cm⁻¹ during electronic relaxation from S1 energy level back to ground state through a conical intersection, whereas an adjacent mode at 185 cm⁻¹ is heavily quenched for Z-1. For E-2 the corresponding excited state mode at 232 cm⁻¹ smoothly loses amplitude as the electronic excited state relaxes back to the ground state. I believe this a key experimental observation which heavily supports the claim that for Z-1 the electronic relaxation towards the conical intersection happens ballistically whereas for E-2 this happens with a diffusive motion. Therefore this should be part of the main manuscript and not be relegated to SI. In fact Figure 5 could go to SI as it consists mostly of a cartoon representation of potential energy and does not add a huge amount of information.

ANSWER 01: We agree with the referee and propose to move to the main text the indicated supplementary figures as panel c. and d. in Figure 3. This allows us to maintain Figure 5, which we believe is useful to illustrate (i) the conceptual notion of vibrationally coherent photoreaction, and (ii) the mechanism proposed in the manuscript to be responsible for turning on / off the vibrational coherence via a modification (or engineering) of the system potential energy surfaces via chemical substitution.

Since the former supplementary figures are now in the main paper, we now also comment on the experimental (weak) significance of the 185 cm⁻¹ mode in the figure caption. Hence we propose a new Figure 3 and caption as follows.

Figure 3: Vibrational Coherence Spectroscopy of Compounds Z-1 (a. and c.) and E-2 (b. and d.). a., b. The power spectra of the differential absorption ΔA oscillatory residuals revealed in Figure 2g and 2h are averaged over the 350-470nm probing window and normalized to 1 at the C=C stretch frequency of 1572 or 1574 cm^{-1} . The vertical scale is linear from 0 to 1 and logarithmic above 1. They reveal the frequencies of the modes in which vibrational wave packets are observed upon resonant excitation at 400nm (blue spectra). The same analysis is performed on the data acquired upon off-resonance excitation at 800nm (red spectra). The black spectra correspond to off-resonant excitation of the pure methanol solvent. Stars indicate the vibrational signatures of the solvent or fused silica cuvette. c. The same Fourier analysis of the Z-1 oscillatory residuals is performed now on a sliding, 1-ps long, time window starting at $t_0=0.1$ ps (red spectrum) or $t_0=0.3$ ps (blue spectrum) i.e. respectively before and after impulsive decay. The dominant 80 cm^{-1} mode persists. The 185 cm^{-1} feature instead disappears upon decay to S_0 , but it seems too narrow to represent an oscillatory signal which would last for only 250 fs, which questions its significance and interpretation, especially in the proximity of the very intense 80 cm^{-1} mode. d. Same as c. for E-2, with t_0 starting times ranging from 0.1 to 0.6. The inset displays the semilog plot of the FFT amplitude of the five dominating modes at 232, 456, 651, 736, 1564 cm^{-1} , as a function of t_0 , and the corresponding monoexponential decay fits. While the amplitude of all higher-frequency modes decay on the 600 to 1200 fs time scale, the damping of the 232 cm^{-1} mode is observed to occur on a significantly faster 230 fs time scale, in line with the 300 fs excited state lifetime of E-2.

2) Although Figure 4 conveys a great amount of information regarding the detailed computational analysis performed in this work, it is incredibly difficult to read. It has to be significantly simplified to more synthetically convey the key conclusions being drawn from it in the main manuscript.

ANSWER 02: We have improved the readability of Figure 4 by rearranging its parts and by moving some parts in the Supporting Information. In particular, while a favorable comparison with the available crystallographic data is still mentioned in the main text, the actual data are now reported in Sec. 2 of the Supporting Information. Note, that the new version of Figure 4 now contains also information of the contribution of the out-of-plane pyrroline ring deformation to the S_1 motion which is useful to deal with the criticism responded in ANSWER #6 but also to deal with the criticism of point 1 of reviewer #3 (see ANSWER #9 below). A section (Sec. 7) reporting on such motion has been added to the Supporting Information.

In conclusion, the following parts have been updated or added:

i) Figure 4 of the main text and its legend:

Figure 4: Mechanistic interpretation of the influence of the methyl substitution on C5. **a, b** Structure of the computed S_0 free energy minima of **Z-1** and **E-2** in methanol at room temperature and values of their relevant structural parameters. The values for the corresponding S_1 energy minima are given in parenthesis and the values of the $C9'-C1'=C4-C3$ and $C2'-C1'=C4-C3$ torsional parameters are given in the Newman projections on the left. A comparison with the available observed crystallographic parameters is given in Section 2 of the SI. **c, d** Computed S_1 trajectory of **Z-1** and **E-2** respectively, in methanol solution, illustrated by (top panels) the S_0 , S_1 and S_2 CASPT2//CASSCF/6-31G*/Amber energy profiles, (middle panels) the changes in electronic structure along the trajectories in terms of the fractional positive charge remaining on the pyrroline moiety of the switch, and (bottom panels) the progression along the reaction coordinate described through double bond twisting ($C9'-C1'=C4-C3$, full line), the skeletal double bond stretching (BLA, dashed line) and the out-of-plane deformation of the pyrroline ring (reflected by the $=N-C2-C3-C4-$ dihedral, dashed-dotted line). FC indicates the configurations of panels a and b. Clnt indicates a conical intersection point. Notice the slow progression of **E-2** along both twisting and out-of-plane deformation coordinates.

ii) Figure S2 and its legend in Sec. 2 of the Supporting Information:

Figure S2: Comparison between computed and observed torsional parameters. **a** Values of the C9'-C1'=C4-C5 and C2'-C1'=C4-C3 torsional parameters from the computed and structural data available for **Z-1**. **b** The same data for **E-2**. However notice that, in this case, the computed parameters are compared with a previously reported [?] precursor of **E-2** which is "sterically close" (i.e. a H atom is replaced with an O atom) to **E-2** which could not be crystallized. The computed ground state values, also given in Fig. 4 of the main text, are in square brackets.

iii) Main text, 8th paragraph:

" Firstly, both crystallographic (see Sec. 2 in the SI) and computed room-temperature structures in methanol solution (see Figures 4a, 4b and Sec. 3 in the SI) show that the reactive C1'=C4 bond is pre-twisted in **Z-1** but substantially planar in **E-2**."

iv) Supporting Information, Sec. 7.

7) Excited state geometrical progression.

Figure S8: Structure of the excited state isomerization coordinate. Left. First 100 fs evolution of the bond length alternation stretching coordinate (BLA), double bond isomerization coordinate ($-C9'-C1'=C4-C5-$) and pyrroline out-of-plane deformation coordinate ($=N-C2-C3-C4-$) along the FC trajectory of **Z-1**. Right. Same data for the FC trajectory of **E-2** trajectories. The times given along the **Z-1** coordinate show that the BLA mode is populated immediately, that the $-C9'-C1'=C4-C5-$ isomerization activate only after 11 fs and that the out of plane deformation becomes active only after 32 fs. In contrast, along the **E-2** coordinate deformation only BLA is active during the first 100 fs.

Figure S9: Contribution of the excited state out-of-plane ring deformations along the isomerization coordinate of Z-1 and E-2. Upper panels. Progression along the reaction coordinate of Z-1 (left) and E-2 (right) of the out-of-plane ring deformation of the indane five-membered ring (described by the C1'-C2'-C3'-C4' dihedral. See dashed-dotted line). The BLA stretching coordinate (dashed line) and -C9'-C1'=C4-C5- isomerization coordinates (full line) are reported for comparison. Lower panel. The same diagram for the out-of-plane ring deformation of the pyrroline ring (reflected by the =N-C2-C3-C4 dihedral, dashed-dotted line). Note, that the coordinate evolution represented in the left and right panel for the pyrroline ring are associated to the left and right diagram of Figure S8. Note also that the out-of-plane motion of the pyrroline ring appears to complete a full oscillation in ca. 250 fs. It can therefore be associated to a frequency of ca. 166 cm^{-1} . The numbering in the represented moiety is consistent with that given in Figure 1 in the main text.

3) *The language with which one is suddenly faced at the start of the conclusion section does not fit with the more general, direct and clear language being used up to then and which makes the paper reading pleasant and understandable. For example, "the S₁ force field of Z-1 transfers the system towards" and "to explain the observed loss of ensemble coherence". If this could be simplified, it could improve understanding of the paper significantly.*

ANSWER 03: We changed (see next-to-the-last paragraph before the Materials and Methods section):

"In conclusion, by using vibrational coherent spectroscopy and quantum chemical simulations, we have shown that the S₁ force field of Z-1 transfers the system towards the Clnt ballistically, with minimal influence of the initial velocities. In contrast, E-2 experiences a more diffusive motion towards the Clnt. The corresponding spreading of the individual decay events would then be enough to explain the observed loss of ensemble coherence."

to:

“In conclusion, by using vibrational coherent spectroscopy and quantum chemical simulations, we have shown that the S_1 force field of **Z-1** triggers a ballistic reactive motion towards the Clnt and decay to S_0 . In a statistical ensemble of molecules in solution at room temperature, the observation of vibrational coherence requires a degree of synchronization between all decay events in the ensemble, which demonstrates the ballistic motion and indicates that such motion is poorly affected by the initial nuclear velocities. In contrast, **E-2** experiences a more diffusive motion towards the Clnt. The corresponding loss of synchronization would then explain the observed loss of ensemble coherence.”

4) In the introduction the authors provide information about the quantum yield of the 11-cis retinal isomerization process, which I believe is a key parameter (and important information), and therefore I was expecting to get know how the biomimetic NAIP chromophore compares. But only in the conclusion this is mentioned. Maybe this could be brought up earlier and then more clearly make the connection of how the key findings in this paper can be used to design better structures with higher quantum yields?

ANSWER 04: We propose to update the introduction in the following way (end of 2nd paragraph):

We changed (see 2nd paragraph):

“For this reason, the proposed coherent S_1 decay of **1** remains hypothetical.”

to:

“For this reason, the proposed coherent S_1 decay of **1** remains hypothetical. In addition, the photoisomerization quantum yield of **1** is significantly lower [13, 14] than that of Rh, and the precise understanding of the S_1 dynamics is an unavoidable prerequisite for the development of more efficient NAIP-based switches.”

Reviewer #2 (Remarks to the Author):

The paper reports a strategy for engineering vibrationally coherent motion in molecular systems that can operate an optomechanical energy transduction. For the study transient absorption measurement and quantum chemical simulations are performed. The topic is highly interesting and the paper is well written.

I have only some questions that should be addressed:

On page 2, second last paragraph: here the reference to “fig. 2 e) and f) “should be given explicitly instead of “Figure 2”.

ANSWER 05: OK, this is done.

Is it clear that resonant and non-resonant excitations will induce the same dynamics in the electronic ground state?

ANSWER 06: This is a very important question that is also closely related to some concern of referee 3. We are grateful to both referees because the raised issue is of central relevance and, regrettably, it was not clearly discussed in the paper.

In order to deal with this issue we now incorporate a new discussion, at the end of the 6th paragraph. Accordingly, we append the paragraph:

“For both compounds **1** and **2**, (...) are not detected in the off-resonant experiments (red spectra, Figure 3a and 3b).”:

with:

This observation is central and may be rationalized as follows. The ISRS mechanism at work here as well as spontaneous Raman scattering may both be described by a wavepacket formalism [REF = Polard et al. Ann Rev Phys Chem 1992; REF = Lee & Heller, J Chem Phys, 1979; REF = Dhar et al. Chem Rev. 1994] which introduces the propagation time τ of the wavepacket on the S_1 PES between the two light field interactions that characterize a Raman transition. When τ is much shorter than a given vibrational period, no evolution occurs on S_1 , such that no Raman activity exists for this mode unless non-Condon effects (i.e. nuclear coordinate dependence of the electronic transition dipole moment) become significant. This conclusion holds for non-resonant Raman processes where $\tau \sim \hbar/\Delta E$ with ΔE the detuning of the light field from resonance [REF = Lee & Heller, J Chem Phys, 1979]] (here, with the 800-nm pump, $\Delta E \sim 1.5\text{eV}$ and $\tau \sim 2.7\text{ fs}$). The same conclusion also holds for resonant ISRS performed with a short enough pump pulse [REF = Tanimura & Mukamel, JOSAB, 1993] since in this case τ is of order of the pump pulse duration (8 fs here). Therefore, we argue that for sufficiently low-frequency modes (i.e. oscillation periods much longer than 8 fs) the vibrational activity induced in S_0 by ISRS is the same for both on- and off-resonance pulses (and results from non-Condon effects, if any). In fact, the analysis of the S_1 reaction coordinate discussed below confirms that only a high frequency stretching mode is activated within 8 fs, while the isomerization motion is activated at a later time.

Also to be able to introduce this new discussion, we change the 3rd paragraph:

“Following a method [20] recently applied to Rh [8], in this letter we use a transient absorption (TA) spectrometer employing sub-8 fs pump pulses to perform vibrational coherence spectroscopy [21] (see Material and Methods, and SI for details). More precisely, we compare the effects of resonant and off-resonance excitations on **1** and **2** where the methyl group on carbon C5 has been replaced by a hydrogen atom (see Figure 1c). While the resonant pulse (400nm, see Figure 1d) creates vibrational wave packets both in S_0 and S_1 , the off-resonance (red to IR) pulse triggers vibrational coherences only in S_0 thus allowing us to isolate the relevant S_1 oscillatory features. As we will detail below such a comparison provides a compelling evidence that:...”

to:

“In this letter we use a transient absorption (TA) spectrometer employing sub-8 fs pump pulses and broadband white light probing to perform vibrational coherence spectroscopy [18] (see Material and Methods, and SI for details), in an experimental approach similar to that [19] recently applied to Rh [9]. When it is resonant with any molecule’s absorption band, such a short pump pulse impulsively produces a non-stationary population described as vibrational wavepacket in S_1 . It may however also produce a vibrational wavepacket in S_0 via the so-called Impulsive Stimulated Raman Scattering (ISRS) process. [20-25] This mechanism still operates with an off-resonant pump pulse which produces vibrational wavepackets in S_0 only. Here, we compare the effects of resonant and off-resonance excitations on **1** and **2** where the methyl group on carbon C5 has been replaced by a hydrogen atom (see Figure 1c). As we will detail below such a comparison provides a compelling evidence that:...”

Figure 1d: Please comment on the non-smooth laser pulse spectra

ANSWER 07: In the Materials and Methods section, we changed:

“The spectrometer may be used with either the 8-fs resonant or 6-fs off-resonant actinic “pump” pulses displayed in Figure 1d. A UV-Vis, spectrally broad (300-900nm) and chirped supercontinuum ...”

to:

“The spectrometer may be used with either the 8-fs resonant or 6-fs off-resonant actinic “pump” pulses displayed in Figure 1d. The off-resonant pulse is produced by spectral broadening of the 1mJ, 40 fs, 800 nm pulse of an amplified titanium-sapphire (Ti:Sa) laser system, via non-linear, guided propagation in a so-called hollow-fiber filled with neon gas. This technique typically produces a structured spectrum (see Figure 1d), which may be recompressed down to 6 fs with conventional chirped mirrors. The on-resonant pulse is produced by broadband sum frequency generation of the latter 6 fs pulse with the fundamental Ti:Sa pulse, resulting in an 8-fs pulse with a similarly structured spectrum. A UV-Vis, spectrally broad (300-900nm) and chirped supercontinuum ...”

It did not become clear to me, also not from the SI, how many trajectories were actually run and evaluated to support the interpretation? Were they performed on CASPT2 level of theory or on CASSCF? In case they were performed on CASSCF some validation with respect to the CASPT2 results should be given.

ANSWER 08:

A single Franck-Condon trajectories was calculated for each isomer starting from the corresponding calculated room temperature equilibrium structure. This type of trajectory are usually assumed to represent the dynamics of the center of the vibrational wavepacket in a sub-picosecond dynamical regime. Therefore, such trajectories provide "average" mechanistic information on the progression of the full population at room temperature. The trajectories were propagated at CASSCF level of theory but the energies were recomputed and rescaled at the CASPT2 and also XMCQDPT2 level of theory to correct the energy profiles. The latter are now reported in Figure S8 of the Supporting Information.

Figure S8: XMCQDPT2 energy profiles. Energy profiles along the CASSCF trajectories of Z-1 and E-2 recomputed at XMCQDPT2 level of theory are shown in (a) and (b) respectively. Note that S_1/S_0 crossing of Z-1 at CASPT2 level is an avoided crossing at XMCQDPT2 level.

The validity of the Franck-Condon trajectories as a mechanistic tool for investigating ultrafast photochemical reactions has been investigated in [Gozem, S.; Melaccio, F.; Valentini, A.; Filatov, M.; Huix-Rotllant, M.; Ferré, N.; Frutos, L. M.; Angeli, C.; Krylov, A. I.; Granovsky, A. A.; Lindh, R.; Olivucci, M. J. Chem. Theory. Comput. 2014, 10, 3074-3084.] [Manathunga, M.; Yang, X.; Luk, H. L.; Gozem, S.; Frutos, L. M.; Valentini, A.; Ferré, N.; Olivucci, M. J. Chem. Theory. Comput. 2016, 12, 839-850.]

We have now improved the description of the above methodological details in the last paragraph of the Materials and Methods section:

"...Excited state PES mapping (see Sec. 5 in the SI) and trajectory calculations have been performed using the CASPT2//CASSCF(12,11)/6-31G*/Amber quantum mechanics/molecular mechanics protocol available through the MOLCAS-Tinker interface [39, 40]. The results of Fig. 4 are confirmed at the XMCQDPT2 level (see Sec. 6 in the SI). FC trajectories, i.e. trajectories that starts from S_0 equilibrium geometries with zero initial velocities, are released from the computationally assigned spectroscopic state (see details in Sec. 4 of the SI). For excited state lifetimes of few hundred femtoseconds, FC trajectories are assumed to represent the motion of the center of the vibrational wavepacket [REF=Gozem et al 2014; REF=Manathunga et al. 2016] and are consistent with the PES scans (see Sec. 5 of the SI)...."

Reviewer #3 (Remarks to the Author):

Gueye et al. studied ultrafast excited-state cis-trans isomerization of MeO-NAIP and deMe-MeO-NAIP in solution by time-resolved absorption spectroscopy employing sub-8 fs pump pulses for pumping. Owing to the ultrashort pump pulses whose duration is shorter than the vibrational periods of molecules, the authors succeeded in observing oscillatory features in the dispersed spectra of the probe pulse, which reflect coherent nuclear wavepacket motion induced with the pump process. Based on the comparison between the data obtained with resonant and non-resonant pump pulses, they found a few vibrations attributable to the excited-state and the ground-state photoproduct, although others are assignable to the vibration of the initial ground state which are induced by the impulsive Raman process. Interestingly, cis MeO-NAIP shows the stimulated emission and photoproduct absorption that indicate ballistic formation of the isomerized photoproduct, and for this molecule, they observed the reaction driven vibrational coherence in the photoproduct. In contrast, for deMe-MeO-NAIP which shows the stimulated emission and transient absorption representing usual populational conversion processes, they only observed excited-state vibrations that are induced by photoexcitation. With help of quantum chemical calculations, they discuss the difference in the ultrafast isomerization process of the two compounds, as well as its relevance to the prototypical ultrafast isomerization of rhodopsin.

Understanding nuclear wavepacket motion in the ultrafast chemical reactions is a key to elucidate the reactive multidimensional potential energy surface and how chemical reactions proceed on it. In particular, the coherent vibration that survives reactions is expected to provide some information about the transition state as well as the conical intersections, so that it is extremely intriguing. Therefore, this has been a central issue in physicochemical studies of chemical reactions for quite long, but it is only recently that it becomes possible to discuss in detail based on the results of ultrafast spectroscopy as well as quantum chemical computation. In this sense, the subject of this study is timely and the result is interesting. As the authors discuss in this paper, isomerization of rhodopsin is certainly a prototype of this problem, but the molecules they picked up are much simpler so that they may be more rigorously examined with quantum chemical calculations.

Because of the interesting problem that this study examined, I feel that this paper may be publishable. However, I have numbers of concerns. Thus, this should be published only if the authors can resolve the following concerns.

1.

The authors compared the data taken with resonant and non-resonant pump pulses for distinguishing the excited-state and ground-state vibrations. It is good, because assignments of numbers of early studies were done in a somewhat careless way, which generated substantial

confusion. However, for the assignment of 80 cm⁻¹ vibration of MeO-NAIP, this is not enough because the selection rules of resonance Raman and non-resonant Raman are different. The former occurs with the Franck-Condon mechanism (A-term mechanism) and the latter obeys the vibronic coupling mechanism (B-term mechanism). Therefore, the authors need to measure spontaneous resonant Raman spectra of the samples and confirm that the 80 cm⁻¹ band does not appear in the steady-state spectra. Recent advances in Raman spectroscopy allows us to readily measure low frequency vibrations, in particular for a mode having such a huge Raman cross section. Even in the case that spontaneous experiments are difficult (e.g. because of strong fluorescence), the discussion on the Franck-Condon activity based on the quantum chemical calculation must be made. It must be easy for the authors because the authors have already the results of computations for both of the S₀ and S₁ states. This is critically important for this paper because all the discussions rely on the assignment of this vibration.

ANSWER 09: This point relates to the question of referee 2 and our answer #6 above. The reviewer is therefore invited to read answer #6 that should clarify his/her point. Here we further comment on the A- and B- terms of resonant versus non resonant SPONTANEOUS Raman scattering and argue that when considering IMPULSIVE resonant Raman scattering, a new characteristic time scale which is the pump pulse duration (much shorter than most vibrational periods and dephasing) enters into play which slightly modifies the picture of the resonant case.

For spontaneous Raman scattering, the Lee & Heller wavepacket description demonstrate that the essential difference between resonant and non-resonant processes reside in the time τ over which the S₁ wavepacket propagates on S₁ in between the two light field interactions that (i) initially projects the ground state wavepacket to S₁ (first interaction), and (ii) project the wavepacket back to S₀ (2nd interaction), after propagation over time τ on S₁. For non resonant interaction, the time-energy uncertainty principle defines the effective τ value, which is extremely short, and therefore the wavepacket has no time to propagate on S₁. If the Condon approximation is valid, the “projections” (actually operated by the electronic state transitions via the transition dipole moment) are “rigorous” projections, such that the wavepacket projected back to the ground state is the same as initially. Hence the initial (e.g. n=0) stationary vibrational wave function is projected on another, orthogonal (n<>0) stationary vibrational state, and the Raman transition probability is zero. Only if the Condon approximation is not valid, the projected wavepackets are distorted, and may no longer be orthogonal to the final state: the Raman transition probability is non zero: this is the contribution of the B-term to off-resonance Raman Spectroscopy.

For resonance spontaneous Raman scattering, the τ propagation time is identified to the vibrational dephasing time (or excited state lifetime if shorter, as in the case of compound **1**) in S₁, and the wavepacket has time to evolve, such that upon projection back to S₀, the Raman signal is non zero even if the Condon approximation holds: this corresponds to the A-term.

However, for the STIMULATED Impulsive Raman Scattering (ISRS), the time evolution in the excited state is of order the PUMP PULSE DURATION, at maximum in the resonant case, since both 1st and 2nd interactions/projections occur within the same pump pulse. Therefore if the pump pulse is very short, such that the wavepacket has no time to evolve in the excited state, the ISRS process, even with a resonant pulse, becomes identical to the non-resonant spontaneous process. This is well illustrated by the work of Tanimura & Mukamel, in which the assumption that the pump pulse is a Dirac (infinitely short) pulse results in the fact that no vibrational wavepacket can be triggered in the ground state by resonant ISRS, unless “non-Condon effects” occur. (i.e. same discussion as for the “B-term”).

In conclusion in the limit of very short pump pulse, the resonant ISRS is expected to be equivalent to the non-resonant (spontaneous or impulsive stimulated) Raman scattering process (rather than to the resonant spontaneous Raman scattering). This limit is more and more valid when considering lower- and lower-frequency modes. Therefore we argue that it is fully verified for the 80 cm⁻¹ in particular. In fact, as already explained in ANSWER #6, the analysis of the computed S₁ reaction

coordinate shows that the only reactive mode that could be possibly activated within Bfs is a high frequency stretching coordinate.

Now, considering spontaneous resonant Raman scattering, based on the above discussion, and according to our computational results which point to a qualitative difference between the excited state evolution of compound **1** and **2**, we expect to see a different low-frequency signature in both compounds, with the signature of the excited state motion along the out-of-plane ring inversion/deformation and torsional coordinate (80 cm^{-1}) possibly present in **1** but not (or much less) in compound **2** where we predict a negligible acceleration along these modes.

Finally, regarding experimental Raman spectroscopy, spontaneous RR spectroscopy was performed on **1** and published in J. Phys. Chem. B, 2014, 118, 12243-12250. Using a double monochromator and resonant excitation (frequency-doubled picosecond Ti:Sa) only vibrational spectra above 1000 cm^{-1} were reported. As far as we are concerned (within our own experimental facility) we were not either able to detect lower-frequency spontaneous Raman scattering (out of the background fluorescence in our case).

2.

I noticed some of the authors' view are too simplified and/or too inaccurate. For example, first, the authors compared the isomerization of MeO-NAIP with that of rhodopsin rigorously, At least for me, however, MeO-NAIP looks a model of 13-cis although the retinal Schiff base of rhodopsin is 11-cis. Second, they discuss the difference between MeO-NAIP and diMe-MeO-NAIP by saying that "minor chemical modifications capable to induce specific electronic effects." However, the authors compared the results of "cis-form" of MeO-NAIP and the "trans-form" of diMe-MeO-NAIP. This is not just a change of methyl substitution but a big difference between the trans and cis forms. The authors must reconsider their discussion and rewrite their statements.

ANSWER 10:

Regarding NAIP "looking like a model of 13-cis" rather than 11-cis is obviously true if one considers the distance from the isomerizing bond to the N atom of the Schiff base function. However, NAIP switches have not been developed to mimic Rh exactly, but rather its PHOTOREACTIVITY. Furthermore, in the present work we focus on a single aspect of this photoreactivity, which is vibrational coherence: a phenomenon that has been observed in rhodopsins holding both the 11-cis as well as the 13-cis PSBR. Thus, we are looking for molecular systems where the S_0 and S_1 PES and CInt driving the isomerization, mimic the S_0 and S_1 PES and CInt of Rh which drives the isomerization of the C11=C12 bond, independently from the chemical similarity between the system and Rh.

To clarify that we do not necessarily focus on reproducing the C11=C12 isomerization of Rh we propose the following modifications:

i) In the 2nd paragraph we now state:

".. As a result, the MeO-NAIP (see structure **1** in Figure 1b) was observed to undergo an ultrafast photoisomerization [14] with transient absorption spectroscopy data displaying low-frequency (60 to 80 cm^{-1} , i.e. $\sim 500\text{ fs}$ period) oscillatory features [15, 16] similar to those of the visual pigment featuring a 11-cis PSBR or light-sensing pigments featuring a 13-cis PSBR [17]...."

and 2nd last paragraph

"... This interpretation leads to the hypothesis that while Z-**1** replicates the coherent dynamics that controls the Rh photochemistry, such a regime is not present in E-**2**, demonstrating that the observed biomimetic behavior can be switched off...."

Regarding the fact that the difference between **1** and **2** is not only the substituent at C5, but also Z versus E, we have now inserted the following modifications:

ii) In the 3rd paragraph, last sentence, we write:

"... ii) an elementary chemical modification of the MeO-NAIP structure, affecting its torsional geometry, quenches the signatures of such critical S₁ motion in **2**."

iii) In the last paragraph before the Materials and Methods section we write:

"... Here we demonstrate that the onset (i.e. turning on or off) of a vibrationally coherent reactive motion can be induced via chemical modification and change in equilibrium geometry...."

iv) In the legend of Figure 1, we write:

" The elementary CH₃ to H substitution in Z-**1** at position C5, alters the Z/E equilibrium in favor of a more stable E form. "

Minor points:

3.

For actual experiments that they performed, they just write "Following a method [20] recently applied to Rh [8] ... (page 2, para 2)" in the main text. It is impossible for readers to figure out what kind of actual measurements they carried out. Addition of relevant description is necessary.

ANSWER 11: We now replace this sentence at the beginning of the 3d paragraph by:

"In this letter we perform UV-VIS transient absorption (TA) experiments with a sub-8 fs pump pulses and broadband white light probing to perform vibrational coherence spectroscopy [18] (see Material and Methods, and SI for details), in an experimental approach similar to that [19] recently applied to Rh [9]."

4.

For explaining the spectra they show, the authors write "These data can be interpreted as a time-resolved linear absorption experiment ... (page 2, para 3)". They might be able to say so if the spectra only exhibit the excited-state transient spectra but, in this case, the stimulated emission, ground-state bleaching and the absorption of the photoproduct appear in the transient spectra. Therefore, such a too-simplified description will confuse the readers, although the authors may have tried to explain their observation to non-experts in a very easy manner. This part must be rewritten, or deleted.

ANSWER 12:

While we disagree with the referee (see the detailed explanation below) we conclude that the original text is confusing.

First, we explicitly refer to W. T. Pollard, S. Y. Lee and R. A. Mathies, The Journal of Chemical Physics, 1990, 92, 4012-4029. This work defines an effective linear susceptibility (from which all linear optical properties may be extracted, such as linear refraction index, absorption, dichroism, et...) which includes the effective contribution of all the terms to be considered in the general 3^d order susceptibility describing a pump-probe experiment in the so-called "sequential scheme" (meaning the pump probe beams do not overlap temporally, such that the early signal resulting from the coherent

interaction between pump and probe beams, sometimes called “coherent artifact”, is disregarded, see e.g. the “doorway-window” description e.g. in *Principles of nonlinear optical spectroscopy*, S. Mukamel, Oxford University Press on Demand, 1999introduce).

Already in the abstract of this paper by Pollard et al., one reads: “A useful simplification is achieved by considering the absorption of the probe pulse as the first-order spectroscopy of the nonstationary state created by the pump pulse”. They actually describe a 3-state model and derive the contribution to this effective linear susceptibility of all possible transitions between these states, which include stimulated emission, excited state absorption, and ground state bleach. Within the same formalism, the photoproduct absorption obviously arise when a photoreaction produces a new electronic state with a transition dipole to a higher lying state.

Second, we note that an even more intuitive way of understanding that photoproduct absorption, excited state absorption and even stimulated have an obvious connection to linear absorption (of non-stationary i.e. non ground states) is to note that:

1) the extinction coefficient for absorption of a molecule is computed at the first order of the perturbation theory (i.e. linear response): the same treatment may be formally developed from any non-ground electronic state.

2) The B Einstein coefficient for (stimulated) absorption and stimulated emission is the same, which means that stimulated emission is also part of the 1st order perturbative response of any NON-ground state electronic state interacting with light, and possessing a non-zero transition dipole moment with a lower-lying state. (Corresponds to a negative extinction coefficient).

Therefore we now suggest a modification of our text with the hope it becomes clearer and no longer confuses the reader about this well establish description of pump-probe spectroscopy in the so-called sequential and impulsive regime. Hence, we change (see beginning of the 4th paragraph):

“The TA data obtained upon resonant excitation of **1** and **2** are displayed in Figure 2a and 2b, respectively. These data can be interpreted as a time-resolved linear absorption experiment performed on the non-stationary states produced both in S_1 and S_0 by the pump pulse [20, 22]. They simultaneously reveal the electronic population decay and the vibrational dynamics in terms of overlapping oscillations.”

to:

“They may be interpreted, via the introduction of an effective linear susceptibility, {Pollard, 1990 #429} as the time-dependent, linear absorption of the probe beam by the non-stationary states impulsively produced by the pump pulse in S_0 and S_1 . Accordingly, they reveal simultaneously transient species population kinetics in the form of UV-Vis absorption (from S_1 , bleached S_0 , or photoproduct) or emission (from S_1 only), as well as the accompanying vibrational dynamics in terms of oscillations.”

5.

The vertical axis of the spectra in Figure 3 is very confusing. The authors mix the linear scale and the log scale without any explanation. They have to clearly mention this issue in the main text as well as in the figure caption.

ANSWER 13:

We change in the main text (see 7th paragraph):

“Of central interest is the 80 cm^{-1} mode, which largely dominates the vibrational power spectrum of **1** in the resonant experiment only (2 orders of magnitude more intense than the 1572 cm^{-1} mode).”

to:

“Of central interest here is the 80 cm⁻¹ mode, which largely dominates the vibrational power spectrum of **1** in the resonant experiment only (2 orders of magnitude more intense than the 1572 cm⁻¹ mode; notice the break in vertical scale in Figure 3a and 3b).”

We also change in the caption of Figure 3:

“The power spectra of the differential absorption ΔA oscillatory residuals revealed in Figure 2g and 2h are averaged over the 350-470nm probing window and normalized to 1 at the C=C stretch frequency of 1572 or 1574 cm⁻¹. They reveal...”

to:

“The power spectra of the differential absorption ΔA oscillatory residuals revealed in Figure 2g and 2h are averaged over the 350-470 nm probing window and normalized to 1 at the C=C stretch frequency of 1572 or 1574 cm⁻¹. The vertical scale is linear from 0 to 1 and logarithmic above 1. They reveal ...”

6.

In page 4, para 2, the authors write, “Reaction-induced vibrational coherence were previously observed in a variety of ultrafast photoreactions”. They have to explicitly mention and explain several examples, carefully choosing the literatures. In my opinions, it was “claimed” in a variety of ultrafast photoreactions, but there are not so many which obtained the consensus of the community.

ANSWER 14:

We do agree with this comment of the referee and take the opportunity to clarify the message we want to deliver in this part of the conclusion.

Since a discussion of any existing controversy about the interpretation of previous results in other molecular systems is out of focus here, we do not cite anymore the work of Rosca et al. 2002, (doming mode of the porphyrin upon photolysis of the diatomic molecule in Myoglobin or cytochromes) since this paper elaborates on a controversy about whether the doming mode is a ground state mode triggered by the ultrafast ligand detachment (Zhu et al. Science 1994), or whether it is an excited state mode which is gating the ligand detachment by controlling the motion towards and away from a crossing between excited and ground states (Liebl et al., Nature 1999).

Instead we keep the other examples but note that reaction-induced vibrational coherence is often not proven to relate to the reaction coordinate (like in Rh or triiodide photodissociation) but instead suggested to be transferred to other (possibly “spectator”, or indirectly coupled) vibrational modes.

Hence we change (see beginning of last paragraph before the Materials and Methods section):

“While reaction-induced vibrational coherences were previously observed in a variety of ultrafast photoreactions [27-30], we demonstrated that the onset (i.e. turning on or off) of a vibrationally coherent reactive motion can be induced via chemical modification.”

to:

“Apart from the two seminal examples of vibrationally coherent triiodide photodissociation [ref Banin&Ruhman 1993] and Rh photoisomerization, [ref 5: Wang et al., Science 1994] unambiguous observation of reaction-induced vibrational coherences were reported in other photoreactions such as e. g. ultrafast low- to high- spin transition in an iron II molecular complex [Consani et al Angw. Chem 2009] or ring-opening in a molecular dyad [Schweighofer et al, Sci Report 2015]. However, in the latter two cases, the vibrational coherence was interpreted as being transferred to low-frequency modes not directly related to the reaction coordinate. Here we demonstrate that the onset (i.e. turning on or off)

of a vibrationally coherent reactive motion can be induced via chemical modification and change in equilibrium geometry.”

7.

In discussion part, the authors discussed their observation with the conical intersection. The conical intersection is very important concept but is often misunderstood, in my opinion. In fact, the new term “conical seam” has been introduced because the conical intersection is not one point on the multidimensional potential energy surface. Since the theoretician who knows this issue very well is in the authors, I suggest mentioning this issue in this paper.

ANSWER 15: We agree with the reviewer. The following change has now been implemented in the revised version of the manuscript (see 8th paragraph in the main text):

"... More precisely, the vibrational wavepacket, whose center is assumed to move along the FC trajectory, would decay along a segment of the intersection space containing an infinite number of Cint's [REF=Gozem et al 2013]..."

Reviewer #1 (Remarks to the Author):

I believe the authors addressed fully my previous concerns and improved the understandability of the manuscript significantly. Therefore I fully support publication.

Reviewer #2 (Remarks to the Author):

The authors have addressed all my questions thoroughly. I agree with most of the answers, except for answer 8. here the claim is that one FC-trajectory is usually assumed to represent the dynamics of the center of the vibrational wavepacket in a sub-picosecond dynamical regime. For reference they give is an article, which discusses this point for the example of reduced retinal chromophores. The example fits for the given molecular device discussed in the present paper, but I do not think that this statement can be generalized. From my own experience on quantum dynamics and trajectory calculations I cannot support this statement as a general statement. My suggestion is that the authors should refer to their own work in which they demonstrate that for a model rhodopsins the FC-trajectory approach works well. Apart from that I highly recommend publication of the present work.

Reviewer #3 (Remarks to the Author):

I carefully read the revised manuscript.
I recommend publication in the present form.

Answers to the referees

Reviewer #1 (Remarks to the Author):

I believe the authors addressed fully my previous concerns and improved the understandability of the manuscript significantly. Therefore I fully support publication.

Reviewer #2 (Remarks to the Author):

The authors have addressed all my questions thoroughly. I agree with most of the answers, except for answer 8. Here the claim is that one FC-trajectory is usually assumed to represent the dynamics of the center of the vibrational wavepacket in a sub-picosecond dynamical regime. For reference they give is an article, which discusses this point for the example of reduced retinal chromophores. The example fits for the given molecular device discussed in the present paper, but I do not think that this statement can be generalized. From my own experience on quantum dynamics and trajectory calculations I cannot support this statement as a general statement. My suggestion is that the authors should refer to their own work in which they demonstrate that for a model rhodopsins the FC-trajectory approach works well. Apart from that I highly recommend publication of the present work.

Reviewer #3 (Remarks to the Author):

*I carefully read the revised manuscript.
I recommend publication in the present form.*

ANSWER to referee #2:

We believe this remark refers to the following sentences of the “**Quantum chemical modeling and mechanistic interpretation.**” subsection:

“More precisely, the vibrational wavepacket, whose center is assumed to move along the FC trajectory, would decay along a segment of the intersection space containing an infinite number of Cint's [32].”

and of the “Methods” section:

“For excited state lifetimes of few hundred femtoseconds, FC trajectories are assumed to represent the motion of the center of the vibrational wavepacket, [45, 46] and are consistent with the PES scans (see Supplementary Figure 6).”

We now specify more explicitly that this assumption, supported by the work presented in the 3 references cited, refer to rhodopsin-like systems. Accordingly we have modified the two sentences above in the following way:

“More precisely, the vibrational wavepacket, whose center is assumed to move along the FC trajectory (see Methods), would decay along a segment of the intersection space containing an infinite number of Cint's [32].”

and:

“As argued for other rhodopsin-like model compounds,^{45, 46} and consistently with the PES scans (see Supplementary Figure 6), FC trajectories are assumed to represent the initial (i.e. within few hundred femtoseconds) excited state motion of the center of the vibrational wavepacket.”